# Simulation and Design of a Prism-Type Ultra-Broadband Microwave Absorber Based on Magnetic Powder/Silica Gel Composites

**DOI:** 10.3390/ma15175803

**Published:** 2022-08-23

**Authors:** Qikui Man, Zhenkuang Lei, Xueheng Zhuang, Guoguo Tan, Shiqi Zhu

**Affiliations:** 1Key Laboratory of Magnetic Materials and Devices, Ningbo Institute of Materials Technology & Engineering, Chinese Academy of Sciences, Ningbo 315201, China; 2Zhejiang Province Key Laboratory of Magnetic Materials and Application Technology, CAS Key Laboratory of Magnetic Materials and Devices, Ningbo Institute of Materials Technology & Engineering, Chinese Academy of Sciences, Ningbo 315201, China

**Keywords:** microwave absorption, metamaterial, CST simulation, edge diffraction

## Abstract

Materials that absorb electromagnetic waves over an ultra-wide frequency band have great potential for military and civilian applications. In this study, a square-frustum-type metamaterial structure was designed and prepared using CI/silica gel composites and flake-shaped FeNi/silica gel composites as the filling substrate. The structural parameters of the square frustum were simulated and optimized using CST Studio Suite. The results show that the optimal performance was achieved when the base consisted of 50 vol.% CI/silica gel composites and 25 vol.% FeNi/silica gel composites with a cross-pattern distribution, the square frustum consisted of 50 vol.% CI/silica gel composites, and the total thickness, base thickness, base-edge length, and top-edge lengths were 5, 1.8, 2.5, and 1.5 mm, respectively. This arrangement can effectively absorb frequencies between 1.8 and 40 GHz, realizing ultra-broadband absorption. The excellent absorption performance of the absorber is attributed to multiple quarter-wavelength resonances and edge diffraction effects.

## 1. Introduction

In recent years, there have been rapid developments in electronics, which have a wide range of applications in industry and daily life. However, they also cause electromagnetic pollution that cannot be ignored and is considered a new challenge for information security and physical health [1,2,3]. Materials with favorable absorption properties (i.e., a wide effective absorption bandwidth (EAB), high reflection loss, and low thickness) are a desirable solution to electromagnetic pollution [4,5]. However, it is extremely difficult to combine these properties in a single absorber. Moreover, practical applications, such as military stealth and electromagnetic protection, often require broadband microwave-absorbing materials. Thus, it is also necessary to develop ultra-broadband absorbers [6,7].

Metamaterials are artificial composites with sub-wavelength periodic structures. Notably, their electromagnetic properties have little relation to those of the material itself; instead, they depend on the periodicity, geometric dimensions, and arrangement of the artificial periodic structure [8,9,10,11,12]. Metamaterials can improve upon the performance of traditional wave-absorbing materials in several ways. For example, they are not limited by the “quarter-wavelength” matching theory and can exceed the performance limits of traditional materials.

Tao et al. [13] proposed a metamaterial absorber coupled with a metal plate structure, where a periodic array was loaded with open resonant rings. This arrangement produced strong resonant absorption in the THz band and achieved an absorption efficiency of 96% at 1.6 THz. Chen et al. [14] combined metal triangles and rectangular strips into an X-shaped metamaterial structure. The gaps between the four metal strips, which were loaded with resistance, produced an effective absorption of 5–13 GHz. Yoo et al. [15] developed a polarization-insensitive hexagonal snowflake-like resonant structure by combining an artificial impedance surface and a resistive–capacitive layer. Between 6.79 and 14.96 GHz, the reflection loss was less than −10 dB.

In addition, the periodic structures of metamaterials have more edges than single-layer structures, and a standing wave effect can be produced by selecting dimensions that increase the EAB [16,17,18]. For example, Li et al. [19] constructed a three-layer stepped-structure absorber using flake-shaped carbonyl iron (CI) and achieved an effective absorption of 4–40 GHz when the absorber was 3.7 mm thick. The large bandwidth was attributed to multiple quarter-wavelength resonances and edge diffraction effects.

Zhou et al. designed a two-layer periodic structure using ultrafine iron powder to realize ultra-broadband absorption of electromagnetic energy between 2.64 and 40 GHz. They analyzed the distribution of the electromagnetic field using the electromagnetic simulation software CST Studio Suite, finding that the magnetic field was concentrated at the upper edge of the material at 32.5 GHz [20]. Owing to multiple matching, this structure also showed double absorption peaks, and the EAB was significantly improved. Thus, periodic multilayer structures are an effective means of achieving ultra-broadband absorbers. Moreover, parameter optimization is time-consuming and laborious, but conducting simulations with CST Studio Suite can simplify the optimization of structural parameters [21,22,23]. Furthermore, a field monitor can be added to the simulation to calculate the electric field, magnetic field, energy loss, and energy flow distribution at a given frequency, which can provide a basis for analyzing the loss mechanisms of the absorber [24,25,26].

This study proposes a metamaterial microwave absorber with a square-frustum periodic structure based on a magnetic powder. Experimental samples were produced by casting and demolding, and the optimal dimensions were determined using CST simulations. Thus, a material with ultra-broadband absorption was produced, which can be used for electromagnetic protection, and the benefits of the CST software for developing broadband absorbent materials were demonstrated.

## 2. Materials and Methods

### 2.1. Structure Design

A square-frustum top and flat-base structure were used to develop an absorber with a wide absorption frequency band. The model and parameters of this structural cell unit are shown in Figure 1. The total thickness of the designed structure was 5 mm, and adjustable parameters included the base thickness t_2_, base-edge length X_1_, and square-frustum-edge (top-edge) length X_2_.

### 2.2. Fabrication

CI powder and ball-milled flake-shaped FeNi powder were used as absorbents, and the final optimized powder volume fractions were 50% and 25%, respectively. The SEM images of sphere CI and flake FeNi are shown in Figure 2. As shown in the figure, the carbonyl iron morphology showed a spherical shape with an average particle size of about 3.34 μm, and the FeNi after ball milling showed an obvious flake shape with an average diameter of about 70.98 μm. Liquid silica gel was used as the binder. In the CI/silica gel composites mentioned in this research, the volume fraction of CI was 50%; in the flake FeNi/silica gel composites, the volume fraction of the flake-shaped FeNi was 25%. The matrix of the periodic structure in this study was composed of CI/silica gel composites or flake-shaped FeNi/silica gel composites. CI is a typical magnetic absorber, whereas FeNi mainly provides magnetic loss and has strong electrical loss [27,28]. The mass ratio was calculated from the volume fraction and density, where the densities of the CI, flake-shaped FeNi, and silica gel were 7.8, 7.6, and 1.1 g/cm^3^, respectively. A typical preparation of a periodic structure with CI/silica gel composites as a filler matrix is as follows: The CI powder and silicone gel were weighed according to their mass ratio, put in a beaker, and stirred with a mixer. Then, n-hexane was added as an auxiliary solvent. A custom mold was sprayed with a mold-release agent to prevent adhesion between the silica gel and the corners of the mold. Once the mixture became viscous, it was poured into the mold. The schematic diagram of the mold is shown in Figure 5b, and in the actual experiment, it was printed and molded by a 3D printer. A vacuum device was used to remove the bubbles that remained from the stirring process. Next, the mold and slurry were placed in an oven and cured at 120 °C for 15 min. Finally, the samples were left at room temperature (~25 °C) for 2 h.

## 3. Results and Discussion

### 3.1. Characterization and Simulation

The morphology of the magnetic powder was characterized using scanning electron microscopy (SEM, FEI Quanta FEG 250) at 10 kV. The electromagnetic parameters of the magnetic powder were measured using a vector network analyzer (VNA, Agilent N5234A) over 0.1–40 GHz. Due to the different fixtures used, the electromagnetic parameters from 0.1 to 18 GHz were measured by the coaxial method and from 18 to 40 GHz by the waveguide method. Specifically, the coaxial method involves the following steps: first, prepare the film obtained after the flow calendaring into a ring with an inner diameter of 3.04 mm and an outer diameter of 7 mm by using a ring punching die; then, put the ring vertically into the coaxial connector; finally, input the thickness of the sample to be measured into the vector network, and the electromagnetic parameters of the material are obtained. The waveguide method is similar to the coaxial method test process, the difference being that the film sample needs to be cut into rectangular shapes. Depending on the test band and the size of the film sample, the waveguide clamps are different. Specifically, in the K-band (18–27 GHz), the sample size is 10.668 × 4.318 mm^2^; in the Ka-band (27–40), the sample size is 7.12 mm × 3.556 mm^2^. The reflection loss of the metamaterials was tested by placing the fabricated metamaterial (180 × 180 mm^2^) on a metal plate of an arch-shaped device, with the transmitting and receiving antennas connected by a vector network analyzer (PNA, Agilent N5234A).

In the simulation software, the simulation range was defined as 0.1–40 GHz, and the unit cell was selected as the model. The model was built according to the designed structure, and the corresponding electromagnetic parameters were input into the material library to define the electromagnetic characteristics of the model. Then, the following steps were taken: first, combine with the interface reflection model; then, set up a single port, with only one port set to the perfect electric boundary, and the other port set to open; finally, click “start” for simulation calculation. The reflection loss is given by the S parameter. The software can calculate the magnetic field, electric field, and energy distribution; just add the field monitor and set the scanning frequency point before starting.

### 3.2. Electromagnetic Parameters of the Magnetic Powder and Microwave Absorption Performance of the Single-Layer Structure

The electromagnetic parameters of the magnetic powder/silica gel composites are summarized in Figure 3. Over the measured range of 0.1–40 GHz, the real part of the dielectric constant of CI/silica gel composites was relatively stable and decreased from 17.5 to 16.8, whereas the imaginary part increased from 0.1 to 2.7. At frequencies greater than 8 GHz, the imaginary part of the permeability of CI/silica gel composites exceeded the real part, i.e., tan *δ_m_* was greater than 1, and the magnetic loss was high. The flake-shaped FeNi/silica gel composites had a higher dielectric constant, with the real part decreasing from 131 to 82 over the same measured range. The real and imaginary parts both peaked at 3 GHz. Above 2 GHz, tan *δ_m_* was close to 1, significantly exceeding tan *δ_e_*, indicating that the properties of the flake-shaped FeNi/silica gel composites were dominated by magnetic loss.

The microwave absorption properties of the single-layer structure are shown in Figure 4. The maximum absorption bands of CI/silica gel composites and flake FeNi/silica gel composites were in the low-frequency region. Notably, the CI/silica gel composites’ absorption peak occurred at 2–4 GHz, the thickness range was 2–4 mm, and the absorption intensity exceeded −25 dB. The flake FeNi/silica gel composites’ absorption peaks were concentrated at 0.1–0.5 GHz, and the thickness was 4–6 mm. However, the EAB was narrow, as shown by the absorption performance graph.

### 3.3. Comparison of Simulated and Experimental Absorption Performance of the Periodic Structure

In our experiments, we prepared periodic structures consisting of 50 vol.% CI/silica gel composites as a filling matrix. A schematic diagram of the structure, a physical map, and the measured simulation results are shown in Figure 5. The base thickness, total thickness, base-edge length, and top-edge length were 1.8, 5, 2.5, and 1.0 mm, respectively. Comparing the measured and simulated results revealed that the periodic structure generated double absorption peaks, and that the measured and simulated peaks were in good agreement with no clear offset, thus confirming that the simulation data can be used as a reference.

### 3.4. Simulating Absorption Properties of Metamaterials with Different Parameters

#### 3.4.1. 50 vol.% CI Composite-Filled Square Frustum and Base

Periodic structures with different parameters were simulated using 50 vol.% CI composite-filled square frustum and base structures, and the results are shown in Figure 6. The periodic structure generated double absorption peaks between 0.1 and 40 GHz, the maximum EAB was 36.3 GHz, and the frequency range was 3.7−40 GHz. Figure 6a shows the relationship between the absorption performance and base thickness when the total thickness, base-edge length, and top-edge length were fixed at 5, 2.5, and 1.5 mm, respectively. When the base was 1.8 mm thick, the EAB was 36.3 GHz and the strongest reflection loss was −19.02 dB. Changing the thickness of the base had little effect on the frequency of the absorption peak. Small base-edge lengths X_1_ did not guarantee absorption strengths greater than −10 dB at high frequencies. As the top-edge length X_2_ increased, the double peaks moved to a lower frequency. When X_2_ ≥ 1.75 mm, the absorber could not exceed an absorption loss of −10 dB at high frequencies. Therefore, the values of X_1_ and X_2_ must be controlled within a reasonable range to ensure strong absorption at high frequencies.

Figure 6d shows the effect of the base thickness t_2_ on the EAB and the greatest reflection loss of the absorber. As t_2_ increased from 1.0 to 1.8 mm, the EAB gradually increased; once t_2_ exceeded 1.8 mm, the EAB decreased slightly. Overall, the base thickness had little effect on the EAB and the greatest reflection loss of the absorber.

The base thickness was fixed at 1.8 mm, which provided the maximum absorption bandwidth, and structures with different base-edge lengths X_1_ were simulated. The results are shown in Figure 6e. When the X_1_ was 2.0 mm, the EAB was only 13.6 GHz owing to the poor absorption performance at high frequencies. As the value of X_1_ increased, the EAB increased significantly to a maximum value of 36.3 GHz at 2.5 mm. The effective bandwidths and greatest reflection loss statistics of absorbers with different top-edge lengths X_2_ are shown in Figure 6f. As X_2_ increased, the effective absorption bandwidth first increased and then decreased, and the EAB reached its maximum value when X_2_ was 1.5 mm.

In summary, for a square frustum and base filled with 50 vol.% CI composites, a maximum absorption bandwidth of 36.3 GHz was achieved when the base height, base-edge length, and top-edge length were 1.8, 2.5, and 1.5 mm, respectively.

#### 3.4.2. 50 vol.% CI Composite-Filled Square Frustum and 25 vol.% FeNi Composite-Filled Base

The absorption performance of 50 vol.% CI composites was weak at low frequencies. In contrast, 25 vol.% flake-shaped FeNi composites had an absorption peak in the lower frequency range; thus, they were used as a base to improve the overall absorption performance. The reflection loss curve and performance statistics for the structure are shown in Figure 7. The base-edge and top-edge lengths were fixed at 2.5 and 1.5 mm, respectively, while the base thickness was varied. A base thickness of 1.0 mm was found to produce an effective absorption bandwidth of 37.7 GHz. Increasing the base thickness shifted the high-frequency absorption peaks to higher frequencies and the low-frequency absorption peaks to lower frequencies.

The base thickness was fixed at 1.0 mm—the value which produced the maximum absorption bandwidth—and structures with different base-edge lengths were simulated. The results are shown in Figure 7b,e. Increasing the base-edge length enhanced the absorption intensity at high frequencies, but the lowest frequency at which reflection loss of −10 dB was achieved gradually increased, which reduced the overall absorption bandwidth. The absorber had a maximum absorption bandwidth of 37.7 GHz when the base-edge length was 2.5 mm.

The base thickness and base-edge length were fixed at 1.0 mm and 2.5 mm, respectively. Then, structures with different top-edge lengths were simulated. The results are shown in Figure 7c,f. When the top-edge length was large, only one absorption peak was produced because the high-frequency peak covered the low-frequency peak. This phenomenon is easily explained as follows: When the top-edge length is equal to the base-edge length, the structure is equivalent to a single-layer slab structure, which can only produce absorption peaks at quarter wavelengths. Thus, the high-frequency absorption peak gradually shifts to lower frequencies as the top-edge length increases. The maximum EAB of 37.7 GHz was achieved when the top-edge length was 1.5 mm.

#### 3.4.3. 50 vol.% CI Composite-Filled Square Frustum and Pattern Base

The cross-combination of 50 vol.% CI composite-filled and 25 vol.% flake-shaped FeNi composite-filled bases was investigated. The bottom of the periodic structure is shown in Figure 8, where the yellow-green denotes CI composites and blue denotes flake-shaped FeNi composites. The matrix of the square frustum consisted of CI/silica gel composites.

The absorption performance of the structure was simulated and calculated, and the results are shown in Figure 9. When the total thickness, base-edge length, and top-edge length were fixed at 5.0, 2.5, and 1.5 mm, respectively, changes in the base thickness t_2_ had little effect on the EAB or absorption intensity. When t_2_ ≥ 1.8 mm, the reflection loss intensity of the absorber could exceed −10 dB in the middle-frequency band; hence, the absorption bandwidth decreased. As t_2_ increased, the low-frequency absorption peak shifted to lower frequencies, and the high-frequency absorption peak shifted to higher frequencies, increasing the total bandwidth. The maximum EAB of 38.2 GHz was obtained when the base thickness was 1.8 mm, and the reflection loss intensity reached −23.2 dB at this time.

The base thickness and top-edge length were fixed at 1.8 and 2.5 mm, respectively. Structures with different base-edge lengths were simulated, and the results are shown in Figure 9b,e. When the base-edge length was 2.0 mm, the EAB of the absorber was 14.8 GHz. When the base-edge length exceeded 2.5 mm, the EAB decreased slightly as the base-edge length increased. The maximum EAB of 38.2 GHz was obtained when the base-edge length was 2.5 mm.

Similarly, taking the top-edge length as a variable, when the top-edge length exceeded 1.5 mm, the EAB increased gradually as the top-edge length increased. Moreover, as the top-edge length increased to 2.0 mm, the EAB decreased sharply to 10.5 GHz. When the top-edge length changed, the strongest reflection loss of the periodic structure did not change significantly, and it remained between −20 and −25 dB.

#### 3.4.4. Electromagnetic Wave Loss Mechanism

Reflection loss curves for monolayer absorbers composed of different thicknesses of 50 vol.% CI/silica gel composites are shown in Figure 10a. As the thickness increased, the absorption peak frequency shifted from the C-band to the L-band. Figure 10b shows that the measured thickness t_m_ is in good agreement with the thickness curve calculated using the quarter-wavelength theory. The absorption peak frequency and the thickness of the absorber satisfied the quarter-wavelength theory. The absorption intensity of the single-layer absorber reached −34.9 dB when the thickness was 2.5 mm, and the EAB was only 2.2 GHz.

The energy loss distribution inside the absorber was calculated using a CST simulation. Consider a periodic structure with an EAB of 38.2 GHz as an example, i.e., a structure with a base consisting of CI/silica gel composites and flake-shaped FeNi/silica gel composites, and a square frustum filled with CI/silica gel composites. If the base height, base-edge length, and top-edge length are 1.8, 2.5, and 1.5 mm, respectively, then the energy loss distribution is that shown in Figure 11. The frequency points selected for analysis were 1.0, 2.2, 4.0, 8.5, 20, and 30 GHz, of which 2.2 and 8.5 GHz are absorption peak frequencies. The low-frequency electromagnetic wave loss was concentrated in the FeNi/silica gel composite base. At higher frequencies, stronger electromagnetic wave loss occurred in the CI composite-filled square frustum. Finally, at frequencies of 20 GHz and above, the electromagnetic wave loss was concentrated at the top of the square frustum. Furthermore, the periodic structure concentrated electromagnetic wave losses at the center of the top surface because the transverse volume of the wave vector was not zero; thus, a standing wave effect occurred on the surface, causing energy losses to aggregate towards the center. In terms of the energy distribution, the energy loss at the surface was greater than that inside the structure, indicating that it produced edge effects. Owing to the existence of these edge effects, the energy loss distribution was uneven, and an arched distribution formed with low to high losses radiating from the inside to the outside.

The comparison of the maximum EAB, thickness, and unit construction of metamaterial absorbers reported in previous work and this work can be seen in Table 1. The metamaterial absorber proposed in this work has the largest effective absorption bandwidth and a comparatively simple cell structure, proving that this type of metamaterial absorber has important applications.

The explanation for the efficient absorption of electromagnetic waves over a wide frequency range achieved by the metamaterial absorber in this study is as follows: firstly, the multiple quarter-wavelength resonance effect due to the bilayer structure allows the absorber to obtain double absorption peaks in the tested range, which is a prerequisite for the absorber to achieve broad frequency absorption; secondly, the edge diffraction effect of the periodic unit has been shown to contribute to less reflection of incident waves [19]. In this study, the diffraction effect of the absorber was enhanced by adjusting the structural parameters of the cell, which makes the incident electromagnetic wave interfere with the phase extinction between periodic cells, a condition that once again enhances the electromagnetic wave loss. Based on these two loss effects, the metamaterial absorber is ultimately an efficient absorber of electromagnetic waves in a wide frequency range.

In addition, the insensitivity of the symmetric geometry to the polarization of the incident wave has been demonstrated in previous studies [14,15,29]. The square-frustum-type metamaterial structure proposed in this work clearly belongs to the symmetric geometry, and therefore the metamaterial absorber of this structure is insensitive to polarization.

## 4. Conclusions

In this study, square-frustum periodic structures with ultra-broadband absorption properties were prepared. These structures significantly increased the effective absorption bandwidth through quarter-wavelength and edge diffraction effects. The structural parameters were optimized using CST simulations. The structure provided an effective absorption between 1.8 and 40 GHz when the base consisted of CI/silica gel composites and flake-shaped FeNi/silica gel composites with a cross-pattern distribution, the square frustum consisted of CI/silica gel composites, and the total thickness, base thickness, base-edge length, and top-edge length were 5, 1.8, 2.5, and 1.5 mm, respectively. As the parameters were optimized, the CST simulations were also used to investigate the loss mechanisms of this type of absorbing body and were shown to be a scientific and convenient method of developing broadband absorbent materials.

## Figures and Tables

**Figure 1 materials-15-05803-f001:**
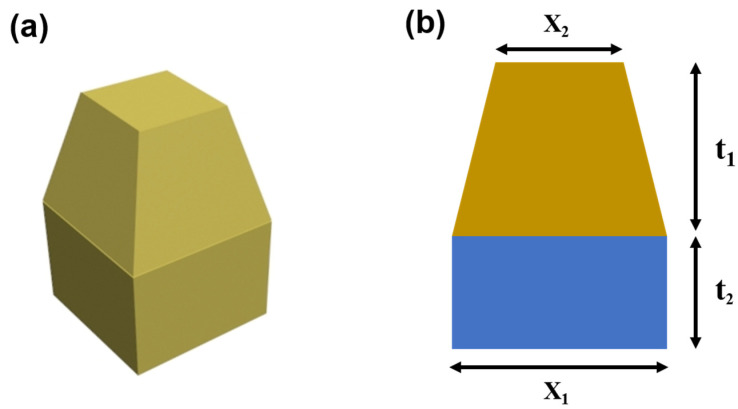
Schematic diagram showing (**a**) the cell structure in this work and (**b**) the parameters.

**Figure 2 materials-15-05803-f002:**
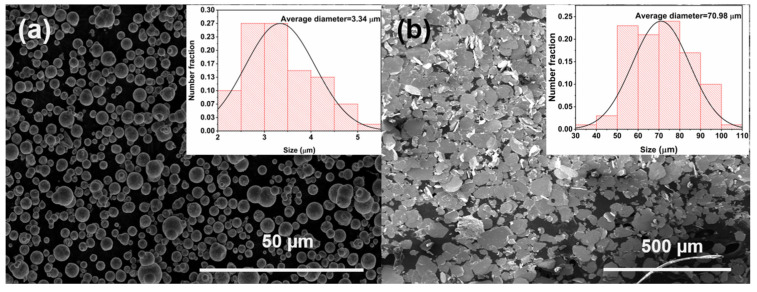
SEM images of (**a**) sphere CI and (**b**) flake FeNi.

**Figure 3 materials-15-05803-f003:**
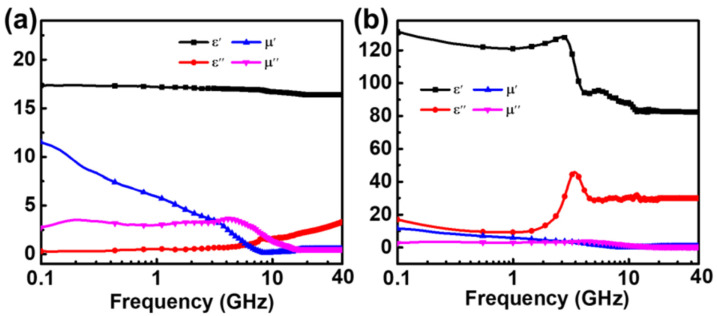
Complex permittivity and permeability of 50 vol.% sphere CI/silica gel composites (**a**) and 25 vol.% flake FeNi/silica gel composites (**b**).

**Figure 4 materials-15-05803-f004:**
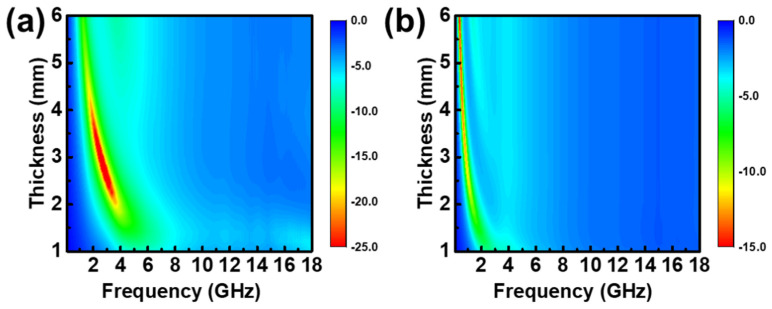
Reflection loss dependence with frequency and thickness of (**a**) 50 vol.% sphere CI/silica gel composites and (**b**) 25 vol.% flake FeNi/silica gel composites.

**Figure 5 materials-15-05803-f005:**
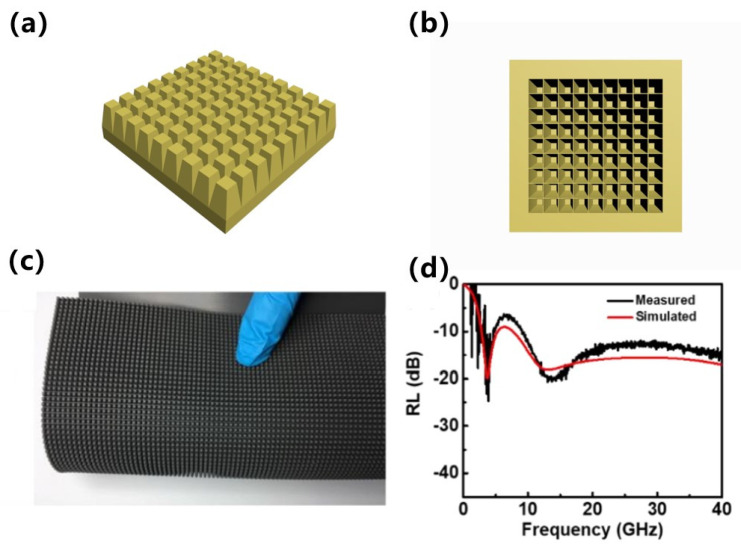
(**a**) Schematic diagram of the model, (**b**) schematic diagram of the model, (**c**) a photograph of the fabricated sample, and (**d**) a comparison of the experimental and simulated results.

**Figure 6 materials-15-05803-f006:**
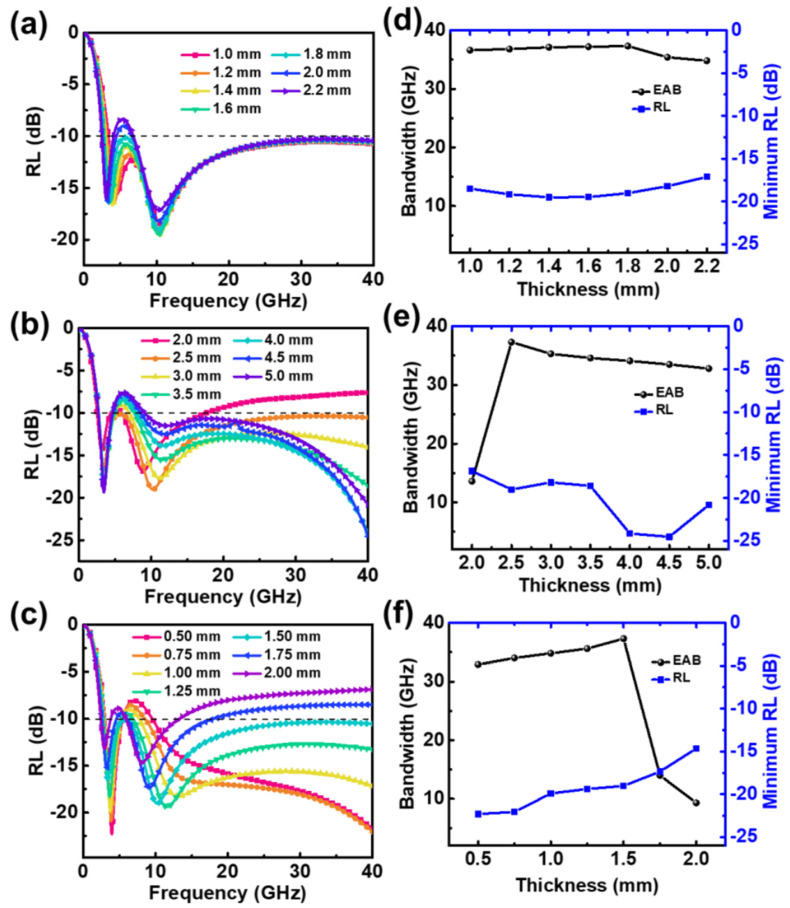
Reflection loss (RL) of absorber with 50 vol.% CI composite-filled square frustum and base against frequency for various (**a**) t_2_, (**b**) X_1_, and (**c**) X_2_. EAB and minimum RL against (**d**) t_2_, (**e**) X_1_, and (**f**) X_2_.

**Figure 7 materials-15-05803-f007:**
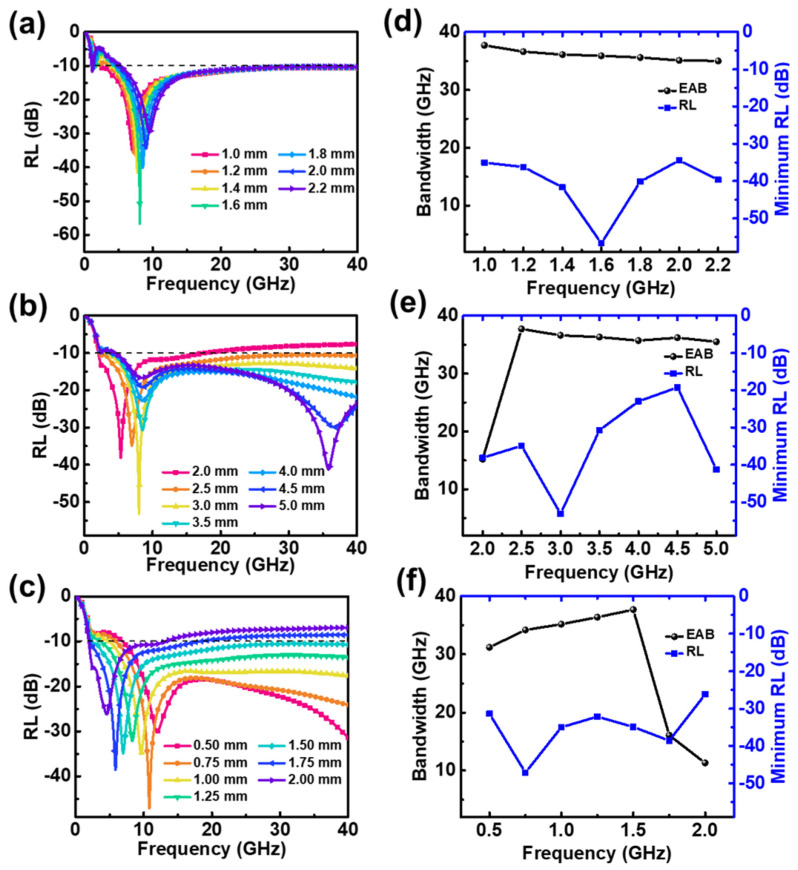
Reflection loss (RL) of absorber with 50 vol.% CI composite-filled square frustum and 25 vol.% FeNi composite-filled base against frequency for various (**a**) t_2_, (**b**) X_1_, and (**c**) X_2_. EAB and minimum RL against (**d**) t_2_, (**e**) X_1_, and (**f**) X_2_.

**Figure 8 materials-15-05803-f008:**
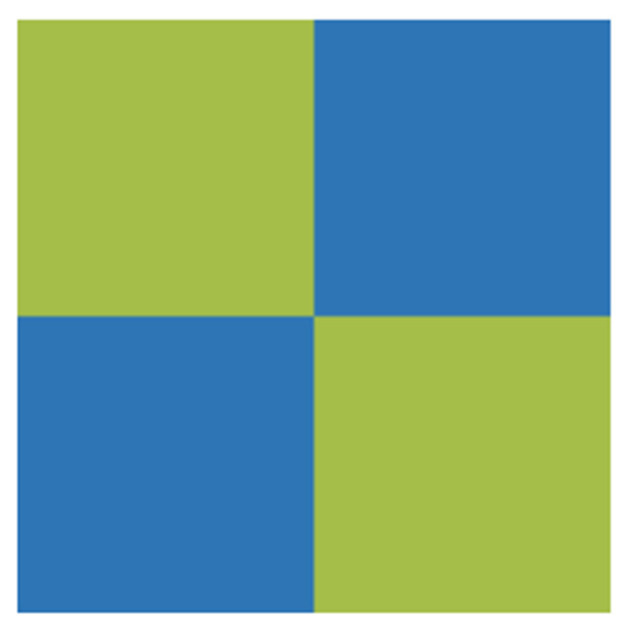
Schematic diagram of the bottom surface of the periodic structure of four units.

**Figure 9 materials-15-05803-f009:**
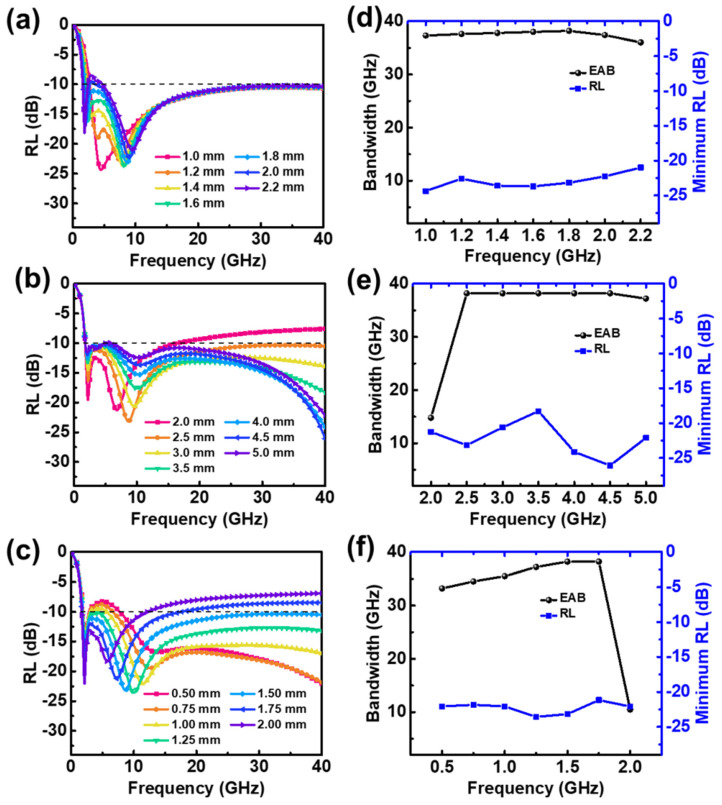
Reflection loss (RL) of absorber with 50 vol.% CI composite-filled square frustum and pattern base against frequency for various (**a**) t_2_, (**b**) X_1_, and (**c**) X_2_. EAB and minimum RL against (**d**) t_2_, (**e**) X_1_, and (**f**) X_2_.

**Figure 10 materials-15-05803-f010:**
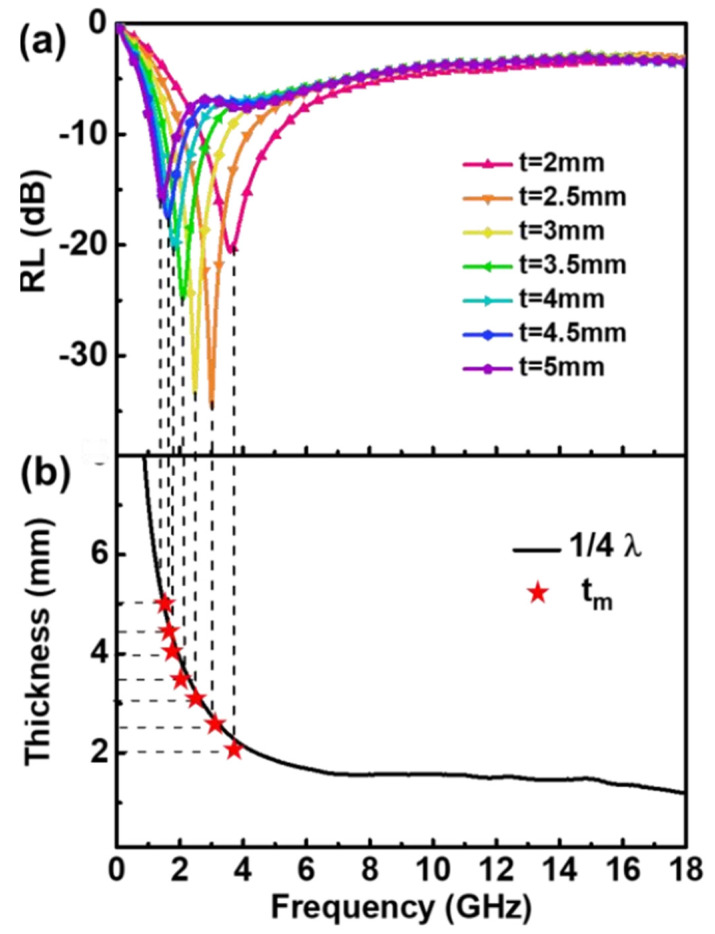
Reflection loss curve of single-layer 50 vol.% sphere CI with different thicknesses (**a**), and dependence of λ/4 absorber thickness on frequency (**b**).

**Figure 11 materials-15-05803-f011:**
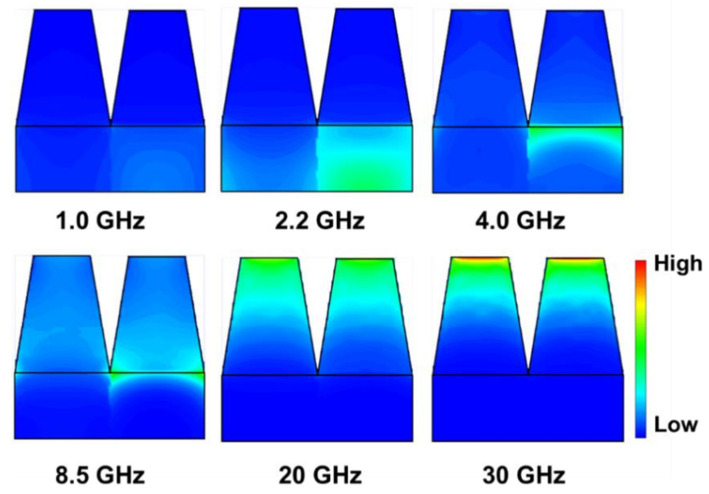
Power loss distribution diagrams of different frequencies: 1.0 GHz, 2.2 GHz, 4.0 GHz, 8.5 GHz, 20 GHz, and 30 GHz.

**Table 1 materials-15-05803-t001:** Comparison of metamaterial absorbers reported in previous work and this work.

Ref	EAB (GHz)	Thickness (mm)	Unit Construction
[14]	8	4	Metal triangle combined with rectangular metal strip
[15]	8.17	4	Hexagonal snowflake shape
[16]	11	4.36	Multi-level pyramid
[17]	34	5	Carbon fiber hierarchical
[18]	16	6	Double steps
[19]	36	3.7	Three steps
[20]	37.36	5.5	Double steps
This work	38.2	5	Prism type

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
