# Peer review of "Simulation and Design of a Prism-Type Ultra-Broadband Microwave Absorber Based on Magnetic Powder/Silica Gel Composites"

_materials, 2022, doi:10.3390/ma15175803_

Round 1

Reviewer 1 Report

The authors designed and optimized the metamaterial absorber in the form of periodic structures with combined fillers and various geometric characteristics of the elements. The results obtained are of high practical importance for the development of broadband electromagnetic radiation absorbers.

There are comments on the submitted manuscript.

So, Figure 1 shows only the shape and parameters (dimensions) of one element of the structure without an image of the relative position of individual elements. Therefore, it is incorrect to talk about the image of the periodic structure in the caption, it appears only in Fig. 4.

In section 2.2, it should be indicated what are the typical particle sizes of magnetic fillers.

Here it is not clearly described how two types of magnetic filler fill the volume of the elements of the structure. At the first reading, it may seem that they are mixed together in a homogeneous way in the material. Only in section 3 comes the understanding that the paper considers various options for the distribution of fillers in the volume of structural elements. In addition, it is not entirely clear what is meant by the matrix. Why silica gel does not belong to the matrix. After all, it is rather a continuous medium in which filler particles (spheres and flakes) are distributed. No information is provided about the substrate on which periodic structures are placed.

Section 3.1 does not describe the details of measurement methods and the equipment used to measure of reflection loss, complex permittivity and permeability. The text does not sufficiently reflect which models and source data were used in the simulation and in what form the samples were in the measurements.

Reviewer 2 Report

In the submitted manuscript materials that absorb electromagnetic waves over a wide frequency band are proposed. The parameters of the absorber were simulated and optimized using CST Studio Suite. The results proposed in the paper are new and should be interesting for a wider audience. However, some additional improvements and explanations are required before the final decision. All comments are listed below.

1.     Authors should explain how the material's characteristics are determined in the wide frequency range.

2.     The potential of the proposed topology should be compared with the characteristic of the other commercial solution and solutions proposed in the literature.

3.     Authors should comment on the setup used in CST for determining the characteristics of the absorber. 

Round 2

Reviewer 2 Report

The authors made significant improvements in the resubmitted manuscript. Most of the reviewers’ comments were addressed. However, the comparison in relation to other solutions proposed in the literature was made only in terms of the EAB parameter, and not the complexity of the construction, dimensions, etc. In addition, the authors did not comment on the influence of polarization on absorption.
